# Capillary pericytes express α-smooth muscle actin, which requires prevention of filamentous-actin depolymerization for detection

Luis Alarcon-Martinez[1,2,3†], Sinem Yilmaz-Ozcan[1†], Muge Yemisci[1,4*], Jesse Schallek[5,6], Kıvılcım Kılıç[1], Alp Can[7], Adriana Di Polo[2,3], Turgay Dalkara[1,4*]

[1]Institute of Neurological Sciences and Psychiatry, Hacettepe University, Ankara, Turkey; [2]Centre de Recherche du Centre Hospitalier de l'Université de Montréal, Université de Montréal, Montréal, Québec, Canada; [3]Department of Neuroscience, Université de Montréal, Montréal, Québec, Canada; [4]Department of Neurology, Faculty of Medicine, Hacettepe University, Ankara, Turkey; [5]Center for Visual Science, University of Rochester, New York, United States; [6]Flaum Eye Institute, University of Rochester, New York, United States; [7]Department of Histology and Embryology, School of Medicine, Ankara University, Ankara, Turkey

**Abstract** Recent evidence suggests that capillary pericytes are contractile and play a crucial role in the regulation of microcirculation. However, failure to detect components of the contractile apparatus in capillary pericytes, most notably α-smooth muscle actin (α-SMA), has questioned these findings. Using strategies that allow rapid filamentous-actin (F-actin) fixation (i.e. snap freeze fixation with methanol at −20°C) or prevent F-actin depolymerization (i.e. with F-actin stabilizing agents), we demonstrate that pericytes on mouse retinal capillaries, including those in intermediate and deeper plexus, express α-SMA. Junctional pericytes were more frequently α-SMA-positive relative to pericytes on linear capillary segments. Intravitreal administration of short interfering RNA (α-SMA-siRNA) suppressed α-SMA expression preferentially in high order branch capillary pericytes, confirming the existence of a smaller pool of α-SMA in distal capillary pericytes that is quickly lost by depolymerization. We conclude that capillary pericytes do express α-SMA, which rapidly depolymerizes during tissue fixation thus evading detection by immunolabeling.
DOI: https://doi.org/10.7554/eLife.34861.001

**\*For correspondence:**
myemisciozkan@gmail.com (MY);
tdalkara@hacettepe.edu.tr (TD)

[†]These authors contributed equally to this work

**Competing interests:** The authors declare that no competing interests exist.

## Introduction

When Rouget first discovered pericytes in 1873, he observed that these cells might serve as contractile entities to regulate microcirculatory blood flow because of their structure and position on microvessels (*Rouget, 1873*). However, this idea was later challenged based on findings that microcirculatory blood flow is largely regulated by upstream arterioles rich in α-smooth muscle actin (α-SMA), the contractile protein present in vascular smooth muscle cells. A number of studies in the past three decades reported that most capillary pericytes in the central nervous system contained little or no α-SMA, whereas α-SMA was detected in pericytes located on pre-capillary arterioles and post-capillary venules (*Bandopadhyay et al., 2001; Nehls and Drenckhahn, 1991; Kur et al., 2012; Kornfield and Newman, 2014; Armulik et al., 2011*). In contrast, recent in vitro and in vivo functional studies have demonstrated that a subgroup of capillary pericytes are contractile in the brain and retina, and have the capacity to regulate the microcirculation by contracting and relaxing in response to physiological (i.e. neurovascular coupling) as well as pharmacological stimuli

**eLife digest** Blood vessels in animals' bodies are highly organized. The large blood vessels from the heart branch to smaller vessels that are spread throughout the tissues. The smallest vessels, the capillaries, allow oxygen and nutrients to pass from the blood to nearby cells in tissues. Some capillaries, including those at the back of the eye (in the retina) and those in the brain, change their diameter in response to activity in the nervous system. This allows more or less oxygen and nutrients to be delivered to match these tissues' demands. However, unlike for larger blood vessels, how capillaries constrict or dilate is debated.

While large vessels are encircled by smooth muscle cells, capillaries are instead surrounded by muscle-like cells called pericytes, and some scientists have suggested that it is these cells that contract to narrow the diameter of a capillary or relax to widen it. However, other researchers have questioned this explanation. This is mostly because several laboratories could not detect the proteins that would be needed for contraction within these pericytes – the most notable of which is a protein called α-smooth muscle actin (or α-SMA for short).

Alarcon-Martinez, Yilmaz-Ozcan et al. hypothesized that the way samples are usually prepared for analysis was causing the α-SMA to be degraded before it could be detected. To test this hypothesis, they used different methods to fix and preserve capillaries and pericytes in samples taken from the retinas of mice. When the tissue samples were immediately frozen with ice-cold methanol instead of a more standard formaldehyde solution, α-SMA could be detected at much higher levels in the capillary pericytes. Treating samples with a toxin called phalloidin, which stabilizes filaments of actin, also made α-SMA more readily visible. When α-SMA was experimentally depleted from the mouse retinas, the capillary pericytes were more affected than the larger blood vessels. This finding supports the idea that the pericytes contain, and rely upon, only a small amount of α-SMA.

Finding α-SMA in capillary pericytes may explain how these small blood vessels can change their diameter. Future experiments will clarify how these pericytes regulate blood flow at the level of individual capillaries, and may give insights into conditions such as stroke, which is caused by reduced blood flow to the brain.

DOI: https://doi.org/10.7554/eLife.34861.002

(*Kornfield and Newman, 2014*; *Hall et al., 2014*; *Peppiatt et al., 2006*; *Fernández-Klett et al., 2010*). Importantly, Hall et al. showed that capillaries covered by pericytes dilate before arte-rioles in response to neuronal stimulus in situ (*Hall et al., 2014*). Recently, an increase in astrocytic calcium was shown to mediate brain capillary changes in response to metabolic demand (*Mishra et al., 2016*). Similarly, Biesecker et al. showed that calcium signaling in Müller cell endfeet was sufficient to evoke capillary dilation in the retina (*Biesecker et al., 2016*). Kisler et al. reported that transgenic mice with a decreased number of pericytes had deficient neurovascular coupling and reduced oxygen delivery to the brain (*Kisler et al., 2017*), providing additional evidence for the importance of pericytes in blood flow regulation. Collectively, these experiments strongly suggest the presence of a bona fide contractile machinery in capillary pericytes. However, a recent paper dismisses the idea of pericytes being contractile by redefining α-SMA-expressing pericytes as smooth muscle cells (*Hill et al., 2015*). To address this paradox between functional and histological findings, we hypothesized that small amounts of α-SMA in capillary pericytes may be rapidly depolymerized during trans-cardial perfusion and fixation thus evading detection by immunolabeling. Consistent with this idea, α-SMA in pericytes has been detected by electron microscopy in which small amounts of α-SMA can be identified (*Toribatake et al., 1997*; *Le Beux and Willemot, 1978*; *Ho, 1985*; *Nakano et al., 2000*) or by in vitro studies in which fixation is more rapidly achieved (*Herman and D'Amore, 1985*). Here, we show that when filamentous-actin (F-actin) depolymerization was prevented by F-actin stabilizing agents or by snap fixation, we detected α-SMA in a much larger fraction of microvascular pericytes, including capillary pericytes placed on the intermediate and deeper retinal vascular beds.

## Results

### Pericytes on retinal capillaries express α-smooth muscle actin

To elucidate the current controversy on the presence of α-SMA in capillary pericytes and to test the hypothesis that low α-SMA immunoreactivity in pericytes could stem from a preparation artifact (*Figure 1*), we first examined F-actin protein expression in capillary pericytes using fluorescently-tagged phalloidin. Intriguingly, we found substantial F-actin in pericyte processes surrounding capillaries (*Figure 1—figure supplement 1A–D*). Encouraged by this observation, we tested several fixation methods to enhance α-SMA detection. Snap freeze fixation of retinas with methanol at −20°C, yielded almost twice as many α-SMA-immunopositive microvessels compared to paraformaldehyde (PFA) fixation (Methanol: 441 ± 28 vessels, PFA: 254 ± 63 vessels, p=0.023; ANOVA) (*Supplementary file 1*; *Figures 1A–B*, *2A, C–D* and *3A–C*). The number of α-SMA-labeled microvessels significantly increased in fourth order capillaries (Methanol: 144 ± 19 vessels, PFA: 60 ± 28 vessels, p=0.003; ANOVA) (*Figures 1A–B*, *2A, C–D* and *3A–C*). Ethanol fixation did not improve α-SMA immunostaining (*Figure 1—figure supplement 2A–B*), suggesting that the effect of methanol was likely due to a faster fixation and not the result of a nonspecific response (e.g. protein denaturation). No immunoreactivity was observed in negative controls when the anti-α-SMA primary antibody was omitted (*Figure 2—figure supplement 1A–B*). In addition to their typical bump-on-a-log appearance detected by lectin, the α-SMA-positive pericytes on capillaries were identified by NG2 or PDGRFβ co-immunolabeling (*Figure 2B–C*), or by their red fluorescence in NG2-DsRed mice, a transgenic line that allows selective visualization of pericytes (*Figure 2D*).

To assess the potential contribution of γ-actin cross-reactivity to α-SMA immunolabeling (*Grant et al., 2017*), especially under fixation conditions that might perturb the equilibrium between different actin isoforms, we used a specific antibody against γ-actin after snap freeze fixation of retinas with methanol. Most of the γ-actin immunostaining ran longitudinally, parallel to the capillary and pericyte plasma membrane, unlike the circular α-SMA outlining the pericyte somata and processes around the capillaries (*Figure 4A–G*). We did not detect any γ-actin immunostaining in pericytes on capillaries after the 4[th] branch. We did not observe a redistribution of the immunostaining patterns of the two actin isoforms when F-actin depolymerization was inhibited with phalloidin (*Figure 4D–G*).

### Prevention of F-actin depolymerization in vivo allows α-SMA detection in capillary pericytes

Phalloidin or jasplakinolide binding to F-actin prevents depolymerization and fixes F-actin in the polymerized state (*Auinger and Small, 2008*; *Cooper, 1987*; *Lee et al., 2010*), thus we reasoned that this strategy might enhance detection of α-SMA in retinal capillary pericytes. To test this, retinas were treated with phalloidin or jasplakinolide followed by fixation with methanol. We found that jasplakinolide or phalloidin treatment (*Figures 1C*, *2B*, *3A–D* and *5A–C*) significantly increased the number of α-SMA-labeling on capillaries of 5[th] and 6[th] order compared to PFA fixation (5[th] order: jasplakinolide = 225 ± 28 vessels, phalloidin = 80 ± 19 vessels, PFA = 15 ± 7 vessels; 6[th] order: jasplakinolide = 123 ± 30 vessels, phalloidin = 23 ± 9 vessels, PFA = 0 ± 0 vessels, p=0.0001, ANOVA) (*Figure 3A*). Besides this treatment revealed additional α-SMA-immunolabeling on 7[th] order capillaries (*Figures 2B* and *5C*) compared to methanol or PFA fixation (jasplakinolide: 31 ± 12 vessels, phalloidin: 20 ± 6 vessels, methanol: 0 ± 0 vessels; PFA: 0 ± 0 vessels, p=0.002; ANOVA) (*Figure 3A*). α-SMA-positive pericytes identified by methanol fixation or phalloidin treatment also expressed NG2 and their number was significantly higher compared to PFA-fixed retinas (PFA: 509 ± 30 vessels, methanol: 883 ± 56 vessels, phalloidin: 890 ± 138 vessels, p=0.035, ANOVA) (*Figure 3B*).

Administration of phalloidin combined with methanol fixation also confirmed abundant α-SMA expression in capillary pericytes of the intermediate (PFA: 97 ± 23 pericytes, methanol: 424 ± 72 pericytes, phalloidin: 509 ± 79 pericytes, jasplakinolide: 497 ± 99 pericytes, p=0.03; ANOVA) and deeper (PFA: 4 ± 2.7 pericytes, methanol: 119 ± 27 pericytes, phalloidin: 260 ± 30 pericytes, jasplakinolide: 359 ± 96 pericytes, p=0.01; ANOVA) retinal plexus (*Figures 2A–D* and *3C–D*, *Figure 2—video 1*). The vast majority of the α-SMA-positive capillary pericytes could only be visualized after methanol fixation at −20°C (*Figures 2A, C–D* and *3C–D*; *Figure 2—video 1*) or upon in vivo

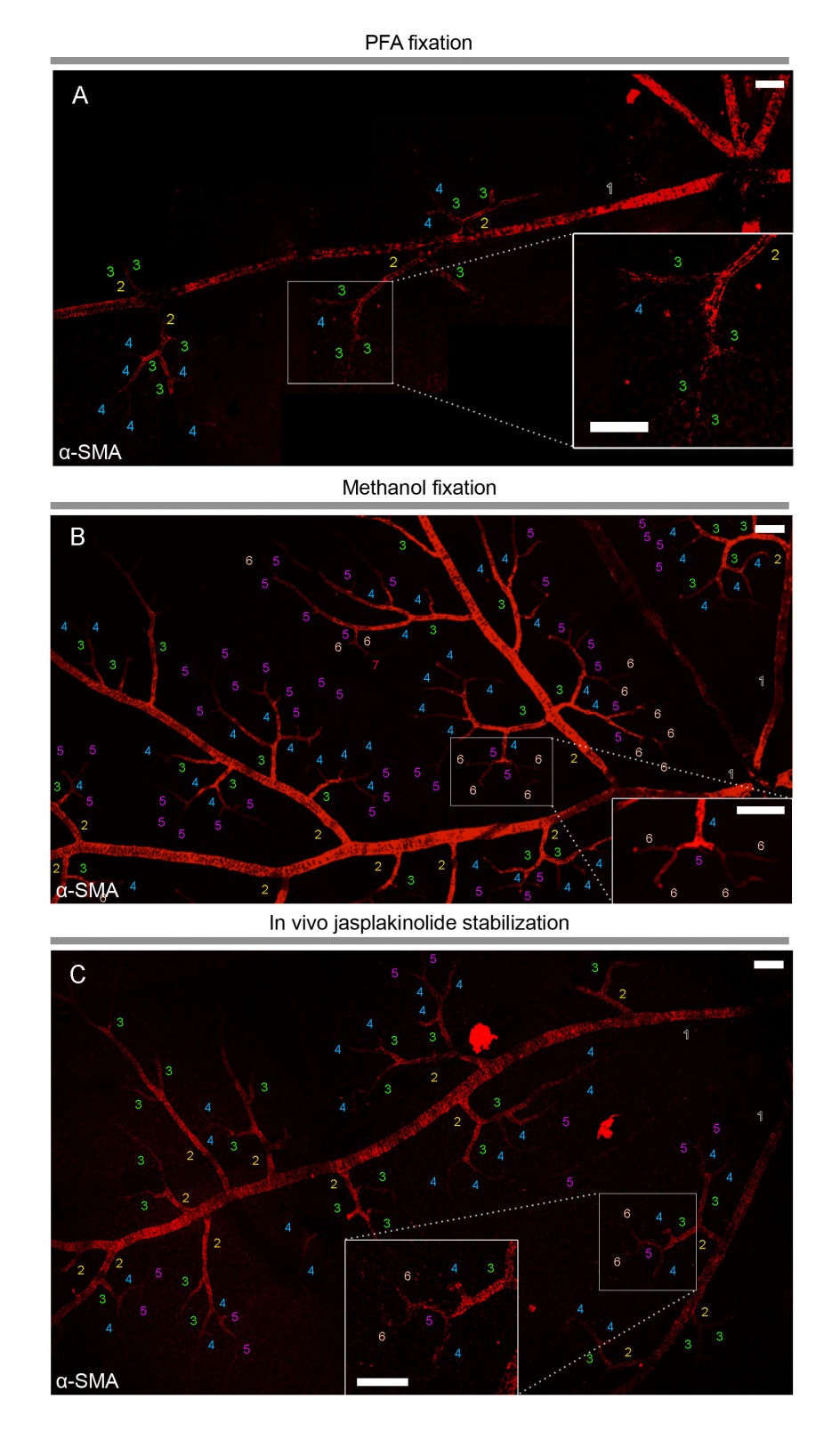

**Figure 1.** α-smooth-muscle actin expression in retinal capillary pericytes can be visualized after rapid tissue fixation or inhibition of actin depolymerization. (**A–C**) The panoramic montage of confocal images of whole–mount retinas illustrate α-SMA immunolabeling of the retinal vessels after PFA (**A**) or −20°C methanol fixation (**B**), or using specific F-actin fixing agent, jasplakinolide (**C**). Each microvessel is numbered corresponding to its branching

*Figure 1 continued on next page*

*Figure 1 continued*

order. Insets show the microvessels in the boxed areas in detail. Faster fixation of retinas with methanol at −20° or inhibition of actin depolymerization shows clear α-SMA immunoreactivity (red fluorescence) in sixth order microvessels, whereas with PFA fixation, α-SMA expression could be visualized only down to fourth order branches. Scale bars, 40 μm.

DOI: https://doi.org/10.7554/eLife.34861.003

The following figure supplements are available for figure 1:

**Figure supplement 1.** F-actin expression in capillary pericytes after PFA fixation.

DOI: https://doi.org/10.7554/eLife.34861.004

**Figure supplement 2.** The effect of methanol was not due to a nonspecific action like protein denaturation.

DOI: https://doi.org/10.7554/eLife.34861.005

administration of jasplakinolide or phalloidin (*Figures 1C*, *2B*, *3A and C–D*). Of interest, we found that pericytes at the junction of two capillaries (i.e. junctional pericytes) were more frequently α-SMA-positive and exhibited a characteristic circular staining pattern wrapping microvessels. In contrast, pericytes on the linear segment of the capillary, which displayed a helical strand-like staining pattern as reported by other groups (*Nehls and Drenckhahn, 1991*; *Hartmann et al., 2015*; *Sims, 1986*), were less often α-SMA-positive.

## Short interfering RNA suppresses α-SMA expression in capillary pericytes

To further confirm α-SMA expression in pericytes, we sought to selectively reduce α-SMA expression using short interfering RNA (siRNA). α-SMA-siRNA significantly suppressed α-SMA expression in capillary pericytes 48 hr after intravitreal administration, while α-SMA expression in pericytes on upstream capillary branches and arterioles was less affected (*Figure 6A–F*). These results are in agreement with the idea that a small pool of α-SMA in capillary pericytes is quickly lost by depolymerization, hence making its histological detection difficult relative to α-SMA-rich pericytes on precapillary arterioles and vascular smooth muscle cells.

## Discussion

In this report, we demonstrate that about 50% of NG2-positive pericytes on high order retinal capillaries (i.e. >5th) located in the intermediate and deeper retinal vascular plexus express α-SMA (*Figure 3D*). Previous reports relied on α-SMA immunohistochemistry involving the slow transcardial infusion of formaldehyde or PFA fixatives (*Thavarajah et al., 2012*), thus lack of or weak α-SMA detection was likely due to rapid F-actin depolymerization (*Huber et al., 2013*) leading to the disruption of the antibody-binding sites (*Dudnakova et al., 2010*). Although antigen retrieval on PFA-fixed retinas revealed some α-SMA labeling in retinal microvessels, this signal was modest and only found in a few capillaries. Here, we show that faster tissue fixation with cold methanol strikingly increased the detection of α-SMA-positive pericytes. It is possible that the detection of the minute pool of α-SMA in small soma and processes of pericytes does not only depend on the fixation method, but might also be difficult to visualize in transgenic mice due to dispersion of the limited amount of membrane-bound reporter protein over the relatively large surface area of the pericyte membrane. Moreover, incomplete (mosaic-like) fluorescent protein expression after tamoxifen injection in inducible transgenic mice, in which Cre recombinase expression is driven by the α-SMA promoter, can also account for the failure to visualize low levels of α-SMA labeling at capillary pericytes (*Hill et al., 2015*; *Hartmann et al., 2015*). Importantly, we demonstrate that inhibition of α-SMA depolymerization in the intact eye using phalloidin or jasplakinolide, two F-actin stabilizing reagents with different pharmacological effects on F-actin, substantially increased visualization of α-SMA in retinal capillary pericytes, particularly those in the deeper retinal plexus. The ratio of α-SMA-positive pericytes was relatively lower to that found in upstream microvessels, consistent with previous reports showing less net $O_2$ delivery from distal capillaries relative to proximal ones (*Sakadžić et al., 2014*). Interestingly, knocking down α-SMA expression led to the disappearance of α-SMA immunostaining mainly in distal capillary pericytes, suggesting that the small pool of α-SMA in capillary pericytes is less stable than in vascular smooth muscle cells. Based on these findings, we conclude

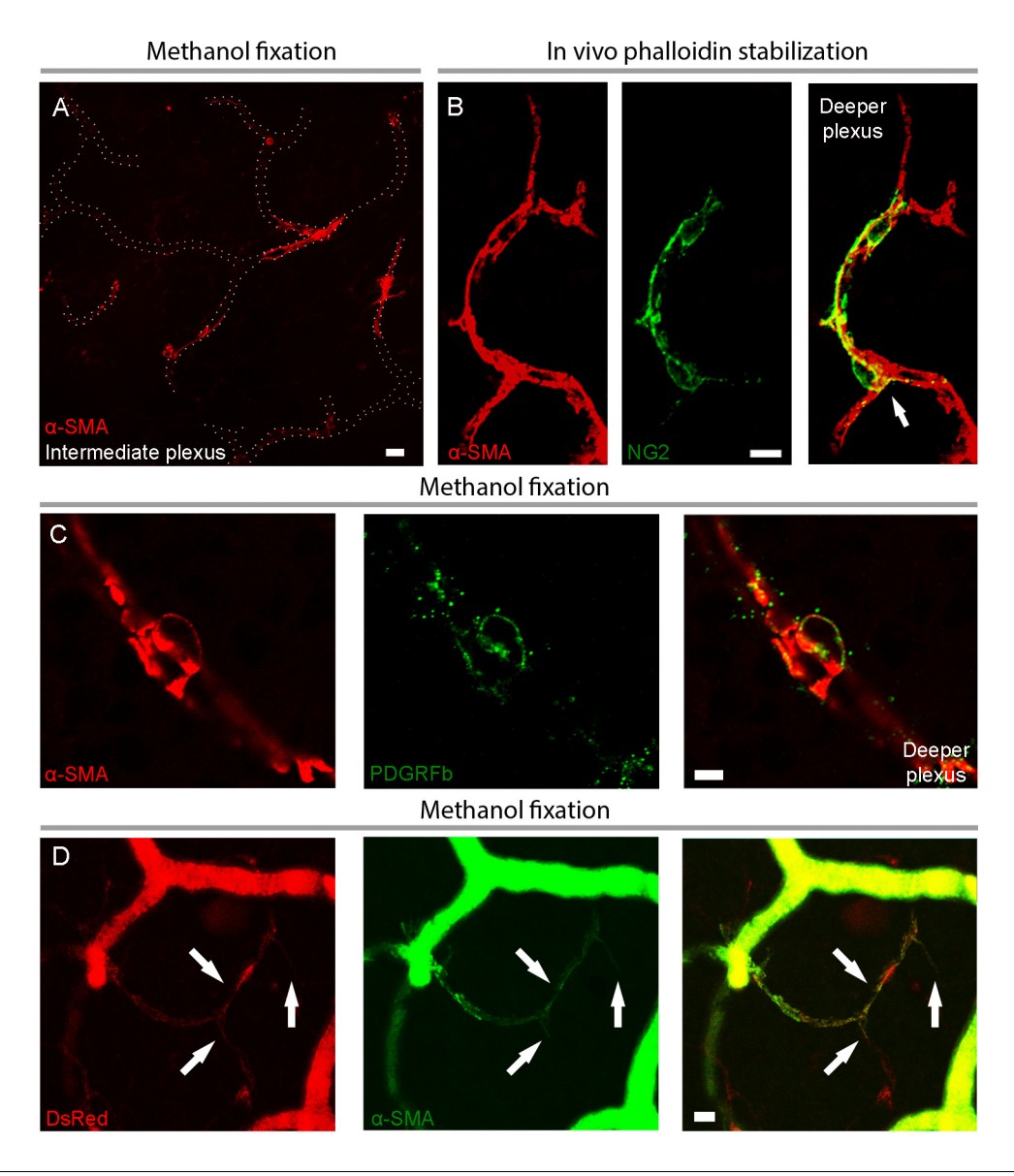

**Figure 2.** Capillary pericytes in intermediate and deeper retinal layers express α-SMA. (**A**) α-SMA expression in the intermediate plexus capillaries after methanol fixation. Note circular pattern of α-SMA staining wrapping a capillary in a junctional pericyte (see the *Figure 2—video 1* for z-stacks of this image). Scale bar: 10 μm. (**B**) Prevention of α-SMA depolymerization in vivo with phalloidin treatment revealed robust α-SMA expression in NG2 positive pericytes also on the deeper plexus capillaries. Note circular α-SMA staining in junctional pericytes, wrapping the capillary wall (arrow). Scale bar: 10 μm. (**C**) Colocalization of α-SMA with PDGFRβ immunoreactivity in a deeper plexus capillary pericyte. Scale bar: 5 μm. **D**) Colocalization of α-SMA immunoreactivity with DsRed fluorescence in several deeper plexus capillary pericytes (arrows) in a retina from an NG2-DsRed transgenic mouse. The focus was adjusted to visualize the deep layer in this image; hence the superficial vessels lack morphological details and appear diffuse. C and D were captured from retinas fixed with methanol. Scale bar: 10 μm.

DOI: https://doi.org/10.7554/eLife.34861.006

The following video and figure supplement are available for figure 2:

**Figure supplement 1.** Negative control for anti-α-SMA immunostaining.
DOI: https://doi.org/10.7554/eLife.34861.007

**Figure 2—video 1.** α-SMA immunostaining in the intermediate plexus capillaries was disclosed after methanol fixation at −20°C.

*Figure 2 continued on next page*

*Figure 2 continued*

DOI: https://doi.org/10.7554/eLife.34861.008

that a large population of pericytes, notably those at branching points of retinal microvessels, have the capacity to express α-SMA, which likely mediates their contraction. It has been suggested that γ-actin might contribute to pericyte contractility in cortical microvessels (*Grant et al., 2017*). However, our results do not support this hypothesis since we did not detect γ-actin-positive capillary pericytes on distal order branches after preventing F-actin depolymerization. The latter finding also indicates that the increase in α-SMA-positivity after phalloidin or jasplakinolide was not caused by cross-reactivity between the antibodies against γ- and α-isoforms or by accumulation of an excess amount of F-actin that might increase the immunostaining of all isoforms.

Unlike peripheral tissues where the blood flow changes homogeneously, the density of pericytes is high in the CNS and retina, where the blood flow demand varies considerably between neighboring cell layers or groups (*Kornfield and Newman, 2014*; *Armulik et al., 2011*; *Schallek et al., 2011*). In vitro studies and recent in vivo brain studies have provided a growing body of evidence that capillary pericytes contract or dilate in response to vasoactive mediators (*Peppiatt et al., 2006*; *Fernández-Klett et al., 2010*; *Puro, 2007*). This blood flow regulation with fine spatial resolution may be essential for tissues with high functional specialization such as the brain and retina. The retinal capillary dilation in response to light stimulus is reportedly layer-specific (*Kornfield and Newman, 2014*). In line with our findings showing clear α-SMA expression in capillary pericytes of the intermediate plexus, the latter study reported robust capillary dilation in this region, but failed to detect α-SMA expression most likely due to the use of PFA-based fixation (*Kornfield and Newman, 2014*). A recent study using transgenic mice expressing fluorescent proteins driven by the NG2 or α-SMA promoters also confirmed the contractile capacity of microvascular pericytes in the cortex in vivo, however, it proposed a radical redefinition by classifying the NG2- and α-SMA-expressing (contractile) cells as smooth muscle cells, rather than pericytes, as they would have conventionally been named under the original Zimmermann definition used since 1923 (*Hill et al., 2015*). The existence of at least three sub-classes of pericytes and transitional forms from smooth muscle cells (*Zimmermann, 1923*) has been a matter of confusion, emphasizing the need for an unambiguous definition of pericyte sub-classes and their corresponding specialized functions.

In conclusion, we identify key components of the contractile machinery in a large population of pericytes in the healthy retina. The identification of α-SMA in capillary pericytes may contribute to clarify the current paradox between functional and histological studies, and expand our understanding of the mechanisms regulating blood flow at the single-capillary level in neurodegenerative conditions including stroke, retinal ischemia, diabetic retinopathy and Alzheimer's disease.

## Materials and methods

### Animals

Seventy three Swiss albino (21–35 g), eleven NG2-DsRed (*Schallek et al., 2013*) mice were housed under diurnal lighting conditions (12 hr darkness and 12 hr light) and fed ad libitum. The number of animals used in each experiment is indicated in the corresponding legend and the *Table 1*, *Supplementary file 1*.

### Study approval

Animal housing, care, and application of experimental procedures were all carried out in accordance with the institutional guidelines and approved by the Hacettepe University Animal Experiments Local Ethics Committee (2012/63), committee guidelines on animal resources at the University of Rochester (Rochester, New York), and the guidelines of the Canadian Council on Animal Care and the Centre de Recherche du Centre Hospitalier de l'Université de Montréal (CRCHUM, Montreal, Quebec, Canada).

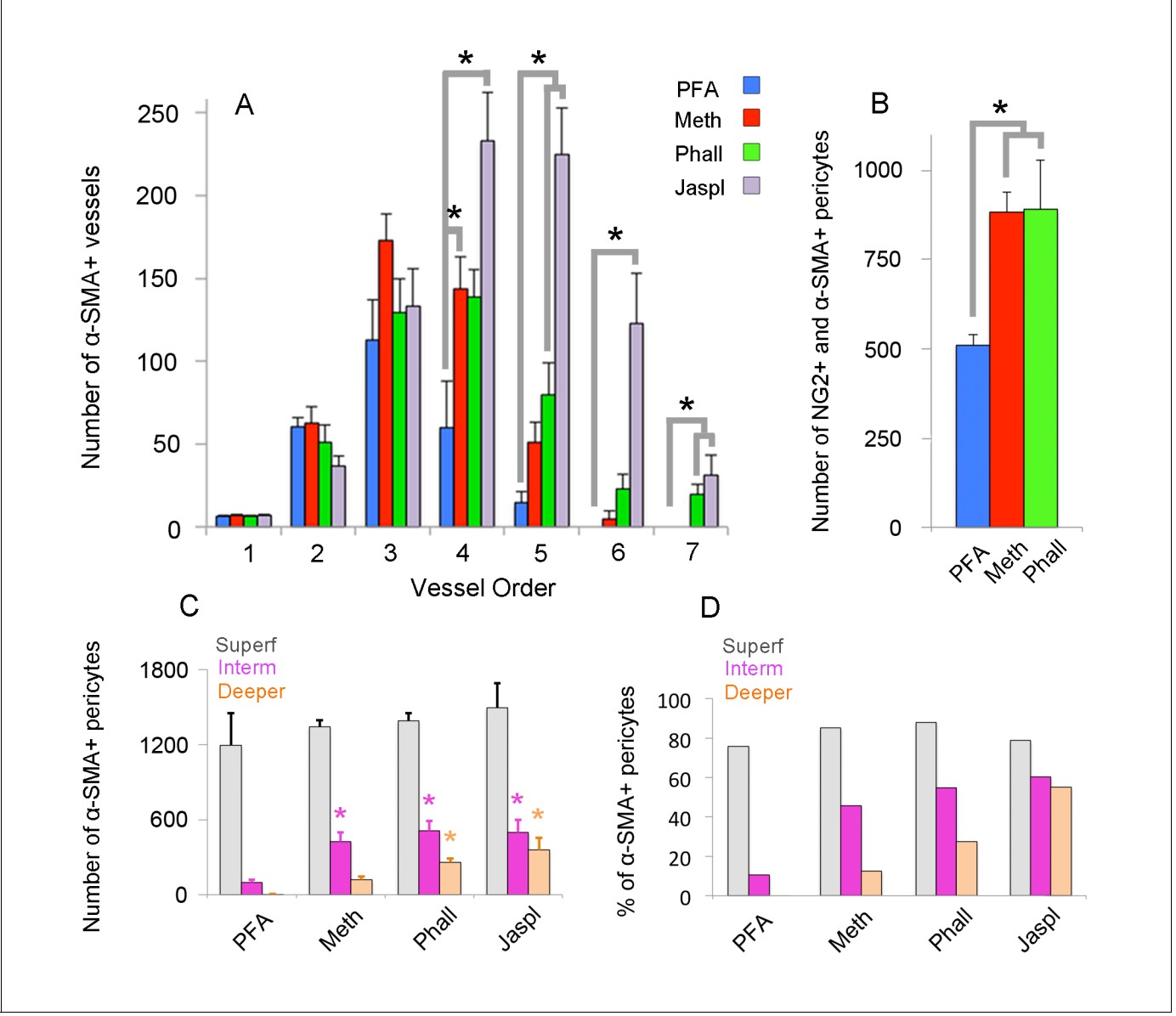

**Figure 3.** Quantification of α-SMA-positive capillary pericytes by vessel order, fixation method and retinal layers. (**A**) Number of α-SMA-positive vessels per retina by the branching order. α-SMA immunostaining was mainly limited to the first four order vessels after PFA fixation (n = 6). In contrast, methanol fixation (n = 5) and phalloidin (n = 5) or jasplakinolide stabilization (n = 3) allowed visualization of α-SMA expression in downstream branches (p<0.05; ANOVA followed by Dunnett's test). Phalloidin and jasplakinolide were especially effective in revealing α-SMA expression in 7th order capillaries, suggesting that their small α-SMA pool rapidly depolymerizes during tissue processing (p=0.002, ANOVA followed by Dunnett's test). (**B**) Total number of NG2 as well as α-SMA + microvessels on each whole-mount retina detected with different fixation methods (PFA: n = 3, 509 ± 30.5 pericytes; methanol: n = 3, 883 ± 56.1 pericytes; phalloidin: n = 3, 890 ± 138.2 pericytes, p=0.035; ANOVA followed by Dunnett's test). (**C**) Illustrates the number of α-SMA-positive pericytes for each retinal vascular plexus. Methanol, phalloidin and jasplakinolide were effective in disclosing α-SMA expression in pericytes located in intermediate plexus (PFA: n = 3, 97 ± 22.7 vs. methanol: n = 3, 424 ± 71.6, phalloidin: n = 3, 509 ± 78.5, jasplakinolide: n = 3, 497 ± 99.5 pericytes respectively, p=0.03; ANOVA followed by Dunnett's test), and especially, phalloidin and jasplakinolide were effective to disclose within the deeper plexus (PFA: n = 3, 4 ± 2.7 vs. methanol: n = 3, 119 ± 27.2, phalloidin: n = 3, 260 ± 29.8, jasplakinolide: n = 3, 359 ± 95.7 pericytes respectively, p=0.01; ANOVA followed by Dunnett's test). (**D**) Illustrates the percentage of α-SMA-positive to total DsRed-NG2-positive pericytes for each retinal vascular plexus in DsRed mice (PFA: n = 3, superficial, intermediate, and deeper: 75.7%, 10.4%, 0.4%; methanol: n = 3, 85.1%, 45.8%, 12.5%; phalloidin: n = 3, 87.9%, 54.9%, 27.3%; jasplakinolide: n = 3, 78.7%, 60.4%, 55.0%). *p≤0.05. (Meth: methanol at −20°C; Phall: Phalloidin, Jaspl: Jasplakinolide, interm: intermediate, superf: superficial).

DOI: https://doi.org/10.7554/eLife.34861.009

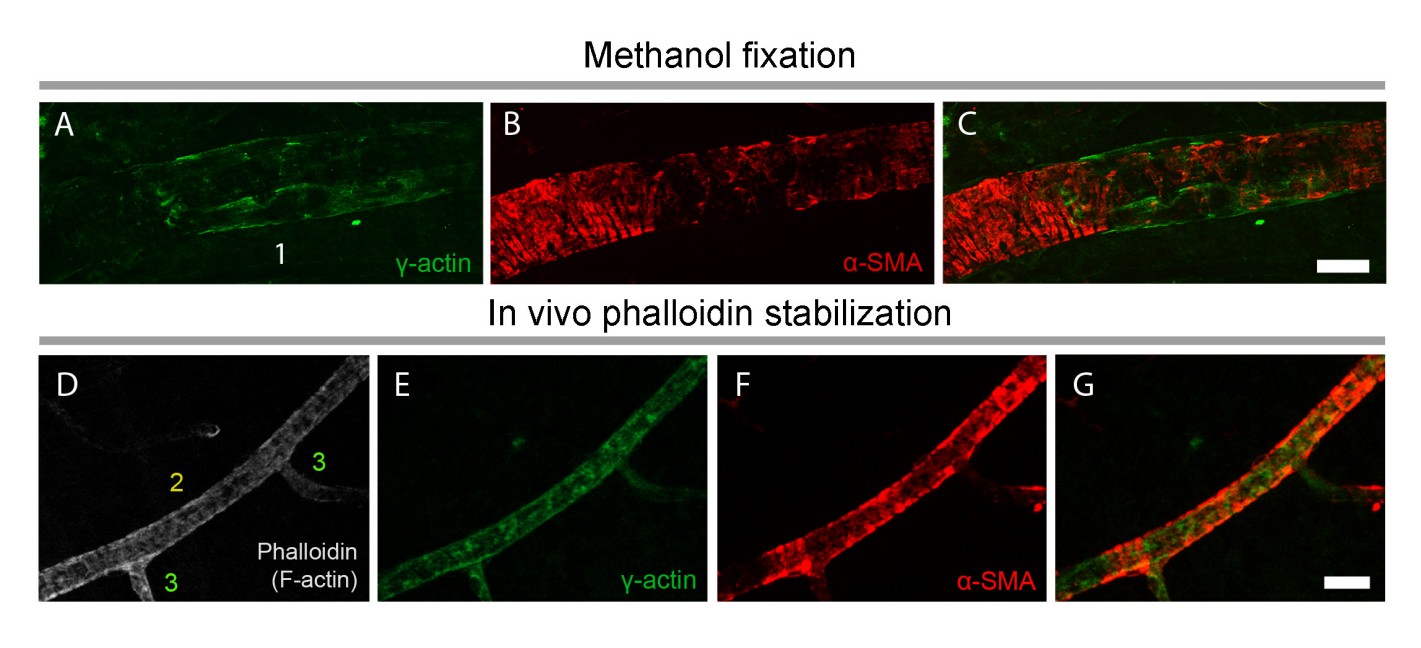

**Figure 4.** γ-actin immunostaining has a distinct pattern than α-SMA immunostaining. (**A–C**) γ-actin immunostaining (green) ran longitudinally parallel to the pericyte plasma membrane unlike α-SMA, which showed a circular immunostaining pattern outlining the pericyte processes around the capillaries (red). F-actin fixation with methanol (**A–C**) or phalloidin (fluorescent-tagged, gray, **D**) did not change γ-actin and α-SMA distribution in vessels (**E–G**). Phalloidin was injected into vitreous 2 hr before sacrificing the mouse. Despite phalloidin stabilization of F-actin filaments, γ-actin remained detectable only ≤4th order branches unlike α-SMA. Scale bars: 20 μm.

DOI: https://doi.org/10.7554/eLife.34861.010

## Retinal immunohistochemistry

Eyes were collected, fixed for 1 hr in 4% PFA at room temperature, and the retinas prepared as flattened whole-mounts by making four radial cuts (*Alarcón-Martínez et al., 2010*). Whole retinas were labeled with lectin (20 μg/ml in PBS containing 0.5% Triton X-100 (PBST), Vector Laboratories, Burlingame, CA) or antibodies against neural glial antigen-2 (NG2) (*Cspg4*) (Millipore, Burllington, MA) and platelet-derived growth factor receptor beta (PDGRFβ) (*Pdgrfb*) (Abcam, Cambridge, UK). Secondary antibody was anti-rabbit IgG conjugated to Cy2 (Jackson ImmunoResearch, West Grove, PA). Briefly, retinas were permeabilized by freezing and thawing in PBST (−80°C for 15 min, room temperature for 15 min), washed 3 times for 10 min, and incubated in 2% PBST at 4°C overnight. The retinas were washed in PBST 3 times for 5 min, incubated in blocking solution (10% fetal bovine serum or normal goat serum in PBST) for 1 hr at room temperature, and then, incubated with each primary antibody diluted in blocking solution (5 μg/ml) at 4°C overnight. The following day, samples were washed in PBST 3 times for 5 min and incubated with secondary antibody diluted in blocking solution (3 μg/ml) for 4 hr at room temperature. We mounted retinas, vitreal side up, on slides and covered them with anti-fade reagent containing Hoechst-33258 to label cellular nuclei (Molecular Probes, Eugene, OR). Retinas were imaged under a light microscope (400x, Eclipse E600, Nikon Instruments Inc., Melville, NY) equipped with a manually controlled specimen stage for X, Y, and Z-axis, a color camera (model DXM1200, Nikon Instruments Inc.), a fluorescent light source (HB-10104AF, Nikon Instruments Inc.), and an image analysis software (NIS-Elements, Version 3.22, Nikon Instruments Inc.). Confocal images of the stained sections were obtained with a Zeiss LSM-510 confocal laser-scanning microscope equipped with a diode laser 488 nm and 561 nm source for fluorescence illumination, and a Leica TCS SP8 DLS (Leica, Wetzlar, Germany) confocal laser-scanning microscope, with a X-, Y-, and Z-movement controller, and a high-resolution PMT (Zeiss, Oberkochen, Germany) and HyD (Leica) detectors. Panoramic pictures of retina were generated by tiling individual images (20x). Samples were visualized with an Apotome fluorescent microscope (Apotome 2, Zeiss) that allowed optical sectioning and was equipped with an automatic controlled specimen

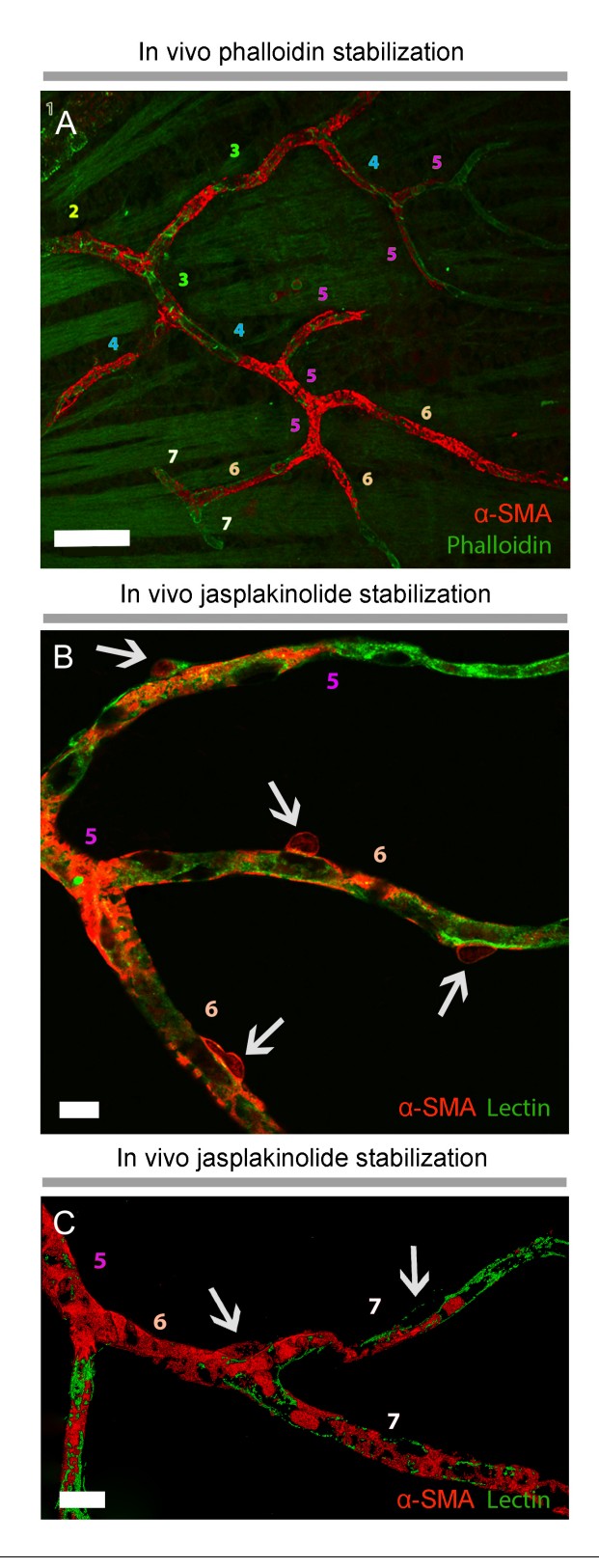

**Figure 5.** Prevention of α-SMA depolymerization in vivo with phalloidin or jasplakinolide revealed further α-SMA immunolabeling in high order retinal capillaries. (**A**), Phalloidin, intravitreally injected for preventing F-actin depolymerization in vivo, was fluorescent-tagged (green) and revealed α-SMA immunolabeling (red) in 6th and 7th order retinal capillaries on whole-mount retinas ex vivo. Scale bar: 40 μm. (**B–C**) F-actin stabilization in vivo with

*Figure 5 continued on next page*

*Figure 5 continued*

Jasplakinolide also disclosed α-SMA immunolabeling (red) in 6[th] and 7[th] order retinal capillaries, which were visualized with lectin (green). Arrows point to pericyte somas and numbers indicate the branch order. Image in C is surface rendered image. Scale bars: 10 μm.

DOI: https://doi.org/10.7554/eLife.34861.011

stage for X, Y, and Z-axis, a color camera (Axiocam 509 mono, Zeiss), a fluorescent LED source (X-cite 120LEDmini, Excelitas, Waltham, MA), and an image analysis software (Zen, Zeiss) for image acquisition.

## α-SMA immunolabeling

After sacrificing the animals, eyes were fixed in PFA at room temperature or −20°C in 100%-methanol for 1 hr. Retinas were collected and permeabilized as described above. Tissue was blocked in

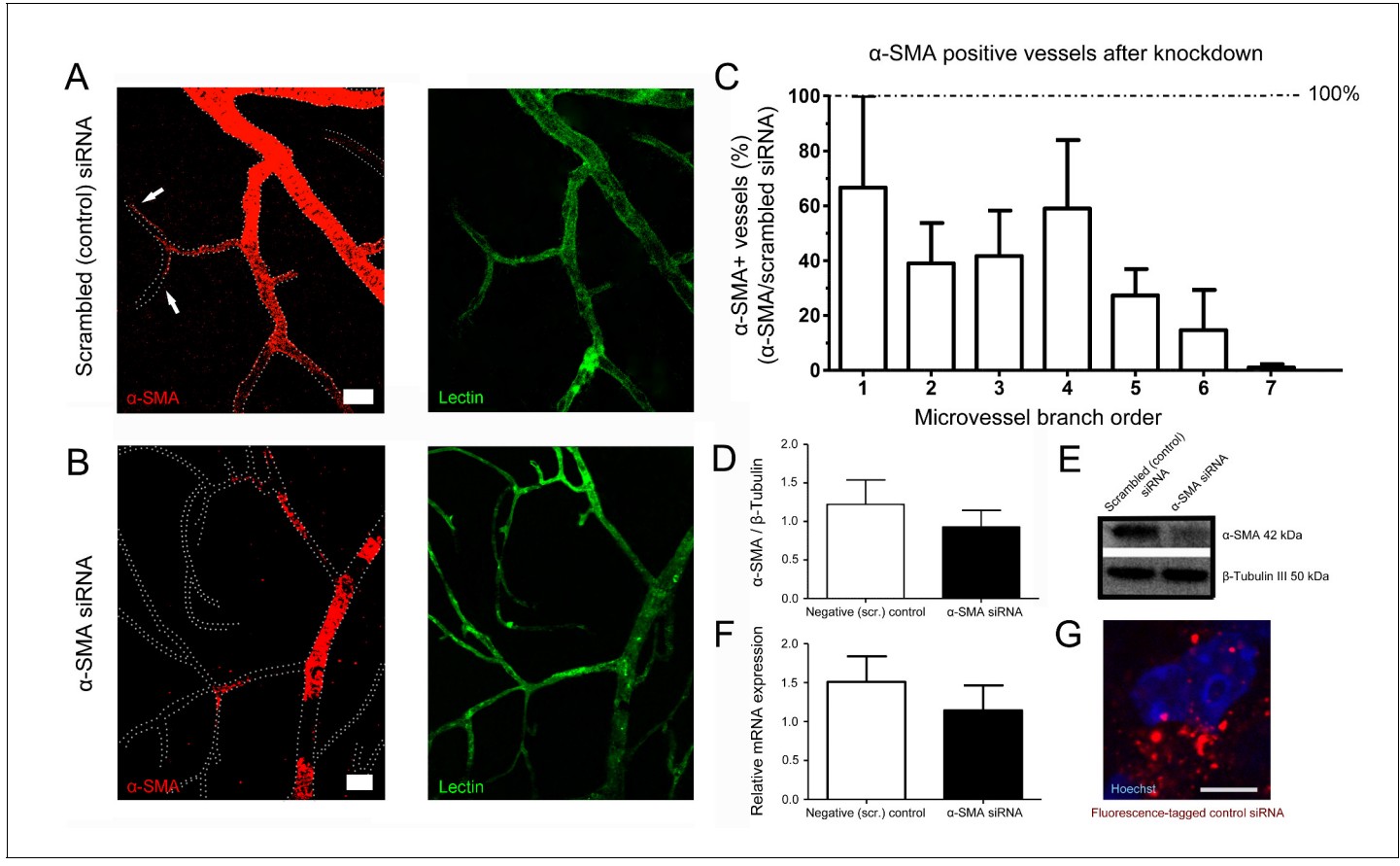

**Figure 6.** α-SMA knockdown by siRNA reduced α-SMA expression. (**A–B**) α-SMA knockdown by siRNA reduced α-SMA expression, being more effective on capillary pericytes (**B**) and only partially effective on more proximal microvessels. Scrambled (control) siRNA had no effect on α-SMA expression (**A**). Note the α-SMA immunostaining extending down to high-order microvessels in **A** (arrows). All retinas were fixed with methanol snap freezing. Scale bars in A, B = 20 μm. (**C**) Illustrates the ratio of microvessels with intact α-SMA immunolabeling in α-SMA-siRNA injected retinas to scrambled (control) siRNA-injected retinas (n = 3 per group). The inhibition trend being more prominent on high order branches was statistically significant (p=0.005) with non-parametric Jonckheere-terpstra test for trend analysis. (**D–F**) Western blotting (**D–E**) (scrambled-siRNA, n = 3, 1.22 ± 0.32 a.u.; α-SMA-siRNA, n = 3, 0.93 ± 0.23 a.u., p=0.4; Student's t-test) and qRT-PCR (**F**) (scrambled-siRNA, n = 3, 1.51 ± 0.33 a.u.; α-SMA-siRNA, n = 3, 1.14 ± 0.32 a.u., p=0.7; Student's t-test) show that both α-SMA protein and α-SMA-mRNA levels were reduced in α-SMA-siRNA injected retinas but the difference did not reach statistical significance possibly because the large amounts of α-SMA in proximal microvessels masked the inhibition in high order branches with small pools of α-SMA. (**G**) Illustrates the fluorescence-tagged control siRNA distributed in the retina 48 hr after injection (red). Hoechst staining identifies the nuclei. Scale bar, 5 μm. (Scr.: scrambled).

DOI: https://doi.org/10.7554/eLife.34861.012

**Table 1.** Agents administered to mice.

The table summarizes all agents injected to mice including company, injection site, volume, concentration, vehicle, and number of mice used.

| Agent | Company | Injection route | Volume | Concentration (Vehicle) | N of mice |
|---|---|---|---|---|---|
| Non-fluorescent phalloidin | Merck Millipore | Intra-vitreous | 2 µl | 5 µg/µl (water) | 4 |
| Fluorescent phalloidin | Biotium | Intra-vitreous | 2 µl | 200 U/ml (water) | 15 |
| Jasplakinolide | Abcam | Intra-vitreous | 2 µl | 10 µM (saline) | 3 |
| Scrambled (control) siRNA | Ambion LifeTech | Intra-vitreous | 3 µl | 0.5 mg/ml (10% Glucose, nuclease free water, in vivo-jetPEI®) | 9 |
| α-SMA-siRNA | Ambion LifeTech | Intra-vitreous | 3 µl | 0.5 mg/ml (10% Glucose, nuclease free water, in vivo-jetPEI®) | 9 |
| BLOCK-iT™ Alexa Fluor® 555 Fluorescent control | Thermo Fisher Scientific | Intra-vitreous | 3 µl | 0.5 mg/ml (10% Glucose, nuclease free water, in vivo-jetPEI®) | 2 |

DOI: https://doi.org/10.7554/eLife.34861.013

10% normal goat serum in PBST at room temperature. Since anti-α-SMA antibodies are commonly generated in mice (*Arnold et al., 2012*; *Taylor et al., 2010*; *Cao et al., 2010*) to avoid non-specific binding to mouse epitopes of the tissue, first we incubated primary antibody against α-smooth muscle actin (α-SMA) (*Acta2*) (Sigma, San Louis, MO) separately with monofragments of the secondary antibody (Jackson Immunoresearch, West Grove, PA) for 2 hr (goat anti-mouse, for one retina: 2 µg of primary antibody with 1.5 µg of secondary antibody in 10 µl PBS). Then, we blocked the potential unbound monofragments by adding 200 µl of 10% normal mouse serum in PBS. Then, tissue was blocked (10% normal goat serum in PBS), and by incubation in the customized primary and secondary antibody mixture overnight at 4°C. Retinas were washed and mounted for visualization as described below.

## Actin depolymerization blockade and analysis of α-SMA-positive pericytes

To prevent actin depolymerization, F-actin was fixed in vivo by 2 µl intravitreal injection of fluorescence phalloidin (200 U/ml, Biotium, Freemont, CA), non-fluorescent phalloidin oleate (5 µg/µl, Millipore, USA), or Jasplakinolide (10 µM, Abcam, UK). Two hrs later, animals were sacrificed and the eyes were collected and fixed in −20°C methanol for 1 hr. Retinas were harvested and subjected to the α-SMA immunostaining protocol described above. Under 200x magnification, we assigned an order number to each vessel segment before branching, starting from arterioles to capillaries, and the total number and the order of α-SMA-positive vessels was determined for each experimental condition. Under 400x magnification and for each fixation method, the number of α-SMA-positive microvessels in each retinal plexus and the total number of α-SMA+/NG2+ pericytes were quantified using a stereological approach. Thus, we analyzed an average of 140 disectors (field of view: 400 × 300 µm along Z-axis) per retina (same area between animal cohorts). The number of α-SMA-positive pericytes in each retinal plexus and the total number of α-SMA+/NG2+ pericytes were calculated using the fractionator equation as follows: total number of elements = Σ quantified elements/ssf X asf X tsf, where ssf is the section-sampling fraction (ssf = number of sections sampled/total sections), asf is the area-sampling fraction (asf = [a(frame)]/area x-y step between disectors), and tsf is the thick-sampling fraction (tsf = frame height/section thickness) (*Leal-Campanario et al., 2017*).

## Short interfering RNA (siRNA) in vivo knockdown

A custom-designed, in vivo specific, HPLC purified α-SMA-siRNA (siRNA directed against *Acta2*) and a scrambled silencer select negative control siRNA were purchased (4457308 and 4404020, respectively, Ambion LifeTech, Carlsbad, CA). This α-SMA-siRNA was previously characterized in wound healing experiments in the murine liver (*Rockey et al., 2013*). Each siRNA was injected into the vitreous using a 34-gauge Hamilton syringe (0.5 mg/ml siRNAs, total volume: 3 µl). Prior to injection,

siRNAs were mixed with a transfection reagent to facilitate cell entry in vivo. Briefly, a transfection mixture composed of 3 µl In vivo-jetPEI (PolyPlus transfection, 201–10G, Illkirch-Graffenstaden, France) and 12.5 µl of 10% glucose in 9.5 µl of nuclease free water was added to the nucleic acid mixture (3.76 µl from 25 µg siRNA, 12.5 µl of 10% Glucose in 8.74 µl of nuclease free water), and incubated for 15 min at room temperature. Transfection mixture was prepared fresh before each knockdown experiment. Forty-eight hrs after intraocular siRNA delivery, mice were sacrificed. To check whether or not siRNAs were taken up by the cells, a fluorescent dye conjugated siRNA (BLOCK-iT™ Alexa Fluor 555 Fluorescent control; Thermo Fisher Scientific, Waltham, MA) was mixed with In vivo-jetPEI and delivered to retina as described above.

Quantitative RT-PCR siRNA-treated retinas were removed precisely under sterile conditions to eliminate RNase contamination. The samples were stored in RNAlater solution (Qiagen, Hilden, Germany, 76104) at −20°C until RNA isolation. RNAs were extracted with RNeasy Mini Kit (Qiagen, 74104) according to instructions. Five hundred ng of total RNA for each sample was used in cDNA synthesis. cDNA synthesis was performed with random hexamer primers with RevertAid First Strand cDNA Synthesis Kit (Thermo Fisher Scientific, K1621) according to instructions. cDNAs were stored at −20°C. Quantitative RT-PCR was performed with Taqman probe-based technology. Taqman gene expression master mix (ABI, Foster city, CA, 4369016), FAM-MGB labeled Taqman probes for mouse α-SMA gene (Assay ID: Mm00725412_s1) and mouse GAPDH gene (Assay ID: Mm9999991_g1) were used. PCR was carried out in triplicates in ABI OneStep Q RT-PCR machine (ABI). Thermal cyclic conditions were as follows: 50°C for 2 min, 95°C for 10 min followed by 40 cycles of 95°C 15 s, 60°C for 1 min. The relative expression values were calculated with $\Delta\Delta$Ct method. α-SMA expression in scrambled-siRNA delivered retinas (n = 3) was normalized to one fold expression and then α-SMA expression in α-SMA-siRNA delivered retinas (n = 3) was compared according to control siRNA group.

Western blot analysis siRNA-treated retinas were removed and protein homogenates were isolated in the presence of a proteinase and phosphatase inhibitor cocktail containing RIPA buffer. Protein concentration was determined by Pierce BCA protein assay kit (Thermo Fisher Scientific, 23225), and 40 µg protein per well was loaded to NuPAGE 4–12% Bis-Tris Protein Gels (Thermo Fisher Scientific, NP0321BOX). Proteins were run and transferred to PVDF membranes, followed by incubation in blocking solution containing 5% BSA solution in TBS containing 0.5% tween-20 (TBST) overnight at 4°C. Blots were incubated with primary α-SMA antibody (2 µg/ml, Sigma, A2547-100UL) diluted in blocking solution for overnight at 4°C then with secondary goat anti-mouse HRP conjugated IgG (0.08 µg/ml, Santa Cruz Biotechnology, Dallas, TX, sc-516102) for 1 hr at room temperature. For loading control, blots were stripped with mild stripping buffer and blocked with 5% fat free milk powder solution in TBST for 1 hr at room temperature, incubated with primary β-tubulin III antibody (0.08 µg/ml, Sigma, USA, T2200) at 4°C for 20 min then with secondary goat anti-rabbit HRP conjugated IgG (0.05 µg/ml, Santa Cruz Biotechnology, sc-2357) for 30 min at room temperature. Bound antibodies were detected with SuperSignal West Femto Maximum Sensitivity Substrate Kit (Thermo Fisher Scientific, 34095). Densitometric analyses were performed with ImageJ software.

After extraction of whole-mount retinas, they were immunostained for α-SMA. Microvessels exhibiting continues α-SMA positivity in the superficial layer of whole mount retinas were counted based on capillary order. Counts were normalized compared to the α-SMA positive microvessel counts from scrambled-siRNA delivered retinas.

## Statistical analysis

All values are provided as the mean ±standard error of the mean (SEM). We evaluated all cohorts with normality (Shapiro-Wilk test) and variance (F-test) tests. For multiple comparisons of values of the stereological quantifications, we used Analysis of Variance (ANOVA) followed by Dunnett's or Tukey's test where appropriate. $p \leq 0.05$ was considered significant. For Western blot and qRT-PCR, two-tailed Student's *t*-test was applied for statistical analysis. To analyze the α-SMA expression relative to capillary order, we used the specific non-parametric Jonckheere-terpstra test for trend analysis.

## Acknowledgements

We thank Prof. D Williams and Prof. B Merigan for allowing us access to laboratory facilities and resources at the Center for Visual Science - University of Rochester, Prof. Y Gürsoy-Özdemir and Dr. B Dönmez-Demir for technical advice and assistance, Prof. M Hayran for his help in statistical analysis.

## Additional information

### Funding

| Funder | Grant reference number | Author |
| --- | --- | --- |
| Seventh Framework Programme | 112C013 | Luis Alarcon-Martinez Turgay Dalkara |
| Türkiye Bilimsel ve Teknolojik Araştirma Kurumu | 112C013 | Luis Alarcon-Martinez Turgay Dalkara |
| Türkiye Bilimsel ve Teknolojik Araştirma Kurumu | 114S190 | Muge Yemisci |
| Research to Prevent Blindness | Career Development Award | Jesse Schallek |
| Research to Prevent Blindness | Unrestricted Grant to the University of Rochester Department of Ophthalmology | Jesse Schallek |
| The Schmitt Program on Integrative Brain Research and NIH | | Jesse Schallek |
| Ruth L. Kirschstein National Research Service Award | NEI F32 EY023496 | Jesse Schallek |
| National Institutes of Health | NEI R01 EY028293 | Jesse Schallek |
| National Institutes of Health | T32 EY007125 | Jesse Schallek |
| National Institutes of Health | P30 EY001319 | Jesse Schallek |
| Canadian Institutes of Health Research | MOP-125966 | Adriana Di Polo |

The funders had no role in study design, data collection and interpretation, or the decision to submit the work for publication

### Author contributions

Luis Alarcon-Martinez, Sinem Yilmaz-Ozcan, Conceptualization, Resources, Data curation, Formal analysis, Supervision, Funding acquisition, Validation, Investigation, Visualization, Methodology, Writing—original draft, Project administration, Writing—review and editing; Muge Yemisci, Turgay Dalkara, Conceptualization, Resources, Formal analysis, Supervision, Funding acquisition, Investigation, Methodology, Writing—original draft, Project administration, Writing—review and editing; Jesse Schallek, Conceptualization, Resources, Data curation, Formal analysis, Supervision, Funding acquisition, Validation, Investigation, Methodology, Writing—original draft, Project administration, Writing—review and editing; Kıvılcım Kılıç, Data curation, Writing—original draft; Alp Can, Data curation, Formal analysis, Supervision, Methodology, Writing—original draft, Writing—review and editing; Adriana Di Polo, Supervision, Methodology, Writing—original draft, Writing—review and editing

### Author ORCIDs

Luis Alarcon-Martinez (iD) http://orcid.org/0000-0002-3167-7423
Muge Yemisci (iD) http://orcid.org/0000-0002-1528-1998
Adriana Di Polo (iD) http://orcid.org/0000-0003-1430-0760
Turgay Dalkara (iD) http://orcid.org/0000-0003-3943-7819

## Ethics

Animal experimentation: Animal housing, care, and application of experimental procedures were all carried out in accordance with the institutional guidelines and approved by the Hacettepe University Animal Experiments Local Ethics Committee (2012/63), committee guidelines on animal resources at the University of Rochester (Rochester, New York), and the guidelines of the Canadian Council on Animal Care and the Centre de Recherche du Centre Hospitalier de l'Université de Montréal (CRCHUM, Montreal, Quebec, Canada).

## Decision letter and Author response

Decision letter https://doi.org/10.7554/eLife.34861.016
Author response https://doi.org/10.7554/eLife.34861.017

## Additional files

### Supplementary files

• Supplementary file 1. Summary of experiments and comparisons. The table summarizes all experiments performed including the treatment groups and number of mice used. Where appropriate, analyses, statistical comparisons, mean ±SEM, and P values are also indicated. Please note that some animals/retinas were used for more than one experiment, therefore, the total number of mice is less than total number of experiments.
DOI: https://doi.org/10.7554/eLife.34861.014

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
