## [Decision Letter]

[Editors’ note: a previous version of this study was rejected after peer review, but the authors submitted for reconsideration. The first decision letter after peer review is shown below.]

Thank you for submitting your work entitled "Pericytes Express α-SMA and Contract During Ischemia due to Failure of Glial Metabolic Support" for consideration by *eLife*. Your article has been reviewed by three peer reviewers, and the evaluation has been overseen by a Senior Editor. The following individuals involved in review of your submission have agreed to reveal their identity: Eric Newman (Reviewer #3). Our decision has been reached after an extensive consultation between the reviewers. Based on these discussions and the individual reviews below, we regret to inform you that the combinations of findings as presented are not suitable for publication in *eLife*.

However, all reviewers thought that the component of the manuscript reporting the expression of SMA in pericytes was a significant contribution after attention to the issues raised by the reviewers on that component of the manuscript. However, all reviewers thought that the other components of the manuscript concerning ischemia, calcium handling and gap junctions would require substantial additional experiments beyond the modest revisions that *eLife* can consider. Thus, we would be willing to consider a resubmitted brief manuscript focused exclusively on the SMA expression components.

If you decide to follow that suggestion, please address the SMA-relevant comments of the reviewers. The comments on the other components of the manuscript are included below for your information. We would have a resubmitted brief manuscript focused on the SMA component re-reviewed by one or more of the original reviewers.

*Reviewer #1*:

This paper is of interest because it addresses a key issue in our understanding of how the CNS regulates its energy supply, i.e. the extent to which capillary pericytes (in addition to arterioles) regulate blood flow. As the authors point out, there has been controversy over whether all pericytes express the α-smooth muscle actin that is assumed to regulate their tone. The authors now seem to resolve this by showing that, with appropriate fixation techniques or phalloidin application, actin can be detected in pericytes on capillaries out to the 7th branching order, i.e. almost all capillaries. The paper also suggests that ischemia induces a [Ca] rise in pericytes and that this calcium influx enters pericytes from glial cells via gap junctions. Although [Ca] is expected to rise in pericytes (indeed in all cells) in ischemia, these two parts of the paper are far less convincing.

Overall, the α-SMA detection part of the paper is a significant result (although, as explained below, it needs better documentation), but the calcium and gap junction parts are not in a state fit for publication. Thus, the paper needs either very significant work and changes to the text, or the latter sections of the text should simply be deleted.

Major points

The authors assume that α-SMA is responsible for pericyte contraction, and it is a major result that using fast methanol fixation they can detect it in pericytes on high branching order capillaries. When the same result is obtained by inhibiting actin depolymerization with phalloidin or jasplakinolide the result is less convincing because these agents may be perturbing an equilibrium between the different types of actin, allowing the formation of a significant amount of F-actin that does not usually exist. This should at least be discussed.

Furthermore, images need to be shown with the primary α-SMA antibody omitted (they are mentioned at the start of the Results section but not shown). Does the antibody cross react with γ-actin, which Grant & Shih (www.ncbi.nlm.nih.gov/pubmed/28933255) have suggested might mediate pericyte contraction?

It is unclear why many of the phenomena reported in the paper are reported using static images rather than by providing a time course. For example in subsection “Retinal ischemia induces α-SMA-dependent sustained pericyte contraction”, is it not possible to provide a graph showing the time course of the reduction of retinal fluorescence and its lack of full recovery?

The measurement of ischemia-evoked constrictions of capillaries and their location near pericytes are poorly documented.a) Larger magnification images are needed for the illustrations in Figure 5C and Supplementary Figure 3. In Supplementary Figure 3 are the black regions of the vessel (i) cells in the lumen, or (ii) occlusions?b) Why are there gaps in the claudin labeling?c) When constrictions are said to localize with pericytes, do the authors mean at pericyte somata (none are visible at some of the constricted areas in Supplementary Figure 3), or within a certain distance of somata (how far?) or do the authors just mean anywhere that they can find NG2 labeling (which may be at most capillary locations!)?d) For the graphs in Figure 5, where were the diameters measured? Was it at pericyte somata or just at randomly chosen locations? For Figure 5, how exactly is a constricted vessel defined, in order to count it? Exactly what criterion was used to choose where to place the arows in Figure 5?e) in paragraph two, subsection “Retinal ischemia induces α-SMA-dependent sustained pericyte contraction”, what length of vessel or area of image is being assessed for the cited numbers of constrictions? Is it the same for non-ischemia and ischemia?f) The statement that constrictions colocalize with helical pericyte processes does not seem to be backed up by an image.g) In subsection “Retinal ischemia induces α-SMA-dependent sustained pericyte contraction”: Give the N numbers for the diameter measurements, say when they were measured precisely, and say whether they were measured at pericyte somata or as an average over capillaries.h) Are the control (pre-ischemia) measurements for phalloidin (4.7 microns diameter) significantly different from the 4.15 microns measured in ischemia? Give exact p values (not just p<0.05) comparing pre-ischemia, ischemia and reperfusion, corrected for multiple comparisons.i) In places constrictions are quantified by their number (a rather subjective judgement – what% reduction in diameter implies a constriction?) and in places by the precise diameter. Please give diameter measurements for the experiments using the siRNA and do a statistical analysis to assess whether they differ from the controls.

Calcium measurements. These are poorly documented. Neither indicator used is ratiometric (and so may be affected – perhaps diminished in intensity – by ischemia-evoked swelling), and the increase in [Ca] is quantified solely as the number of cells with a value above some threshold level. Either provide a time course for the signal, or explain why one cannot be shown.

Carbenoxolone effects.a) The claim that CBX decreases pericyte [Ca] in ischemia would be better shown as a time course of [Ca]i versus time for cells with and without CBX.b) Carbenoxolone is reported to be faintly yellow (Sigma datasheet); does it affect the Ca-sensing dyes used?c) The claim that glial [Ca] rises in CBX needs quantifying properly so that it is not just an anecdote.d) CBX is a famously non-specific drug. Its effects on pericytes may result indirectly from its actions on voltage-gated calcium channels in the retina (www.ncbi.nlm.nih.gov/pubmed/15028741) or on NMDA receptors (www.ncbi.nlm.nih.gov/pubmed/18555495). The suggestion that it prevents Ca influx from glia to pericytes would need far more proof to establish it as plausible.

Gap junction labeling: I am not convinced by this, and the issue might be better studied by injecting dye into glia and measuring whether it spreads to the pericytes. In the insets for Figure 3, the apparent green label, consisting of parallel lines oriented from the pericyte to the nearby glial cell is apparently interpreted as being gap junctions coupling the two cells. I am 99% convinced that this is an artefact. Long drawn out labeling like this is classically seen in confocal images when the fluorescence saturates the photomultiplier (as the green signal appears to do here), causing its output signal to "hang-up" while the scan moves on to other pixels (the fact that all the green "lines" are in the same direction is consistent with this explanation). I would therefore predict that the inset in Figure 3 has been rotated approximately 25 degrees anti-clockwise from the original scan direction, while that in Figure 3 is roughly aligned with the original scan direction. It would in any case be implausible for the connexins forming a junction linking glia and pericytes to be ~ 5 microns long as those in the inset to Figure 3 appear to be. (Alternatively, it is conceivable that the green line axis is the z axis of the original scans, with poorer resolution, and that the image has been rotated for viewing from the side, but in that case I'd expect similar poor resolution for the red and blue images in the same direction, which does not appear to be the case).

Subsection “Mechanisms of ischemia-induced pericyte contraction and pharmacological rescue”, paragraph 2. If the diameter of arterioles can be measured in adenosine, why not measure the diameter of the capillaries near pericytes?

The data with DAB support the idea that glial glycogen can delay the effects of ischemia (though not necessarily by affecting Ca movement through gap junctions), but the effect on the time course described in subsection “in vivo visualization of ischemia-induced pericyte constrictions with AOSLO” is anecdotal. If constrictions appear within 30 mins in the presence of DAB, we need a time course provided for their appearance in the absence of DAB (with a measurement at the same 30 min time point), and some kind of statistical analysis, if a rigorous conclusion is to be reached about whether DAB speeds constriction.

Figure 3. Can the effect of the siRNA expression on SMA expression be quantified as a function of capillary branch order?

*Reviewer #2*:

In their manuscript "Pericytes Express α-SMA and Contract During Ischemia due to Failure of Glial Metabolic Support," the authors Alarcon-Martinez et al. first utilize a new technique to visualize α-SMA in high order capillaries, adding evidence to one side of the field's ongoing debate regarding blood flow control by pericytes in the capillary bed. There is a large debate in the field of cerebrovascular biology as to whether localized blood flow is controlled only at the arteriole level by constriction of vascular smooth muscle cells, or also at the level of capillaries by contraction of pericytes. One of the major arguments against the latter is a lack smooth muscle actin in pericytes. Here the authors use novel techniques to demonstrate that these cells do express smooth muscle actin that is more labile than the smooth muscle cells. This adds a significant advancement to this debate. They then use a model of ischemia to demonstrate how a functional deficit in these microvessel pericyte contractions might affect reperfusion after injury. Lastly, they identify a mechanism by which this deficit might occur. The work is thorough, and the manuscript provides an interesting addition to a current debate. However, there are a few issues that need to be addressed:

1) There is no good evidence in the literature that the siRNA technique used in Figure 3 works. While a labeled siRNA would be more ideal, the knockdown should at least be quantified. Additionally, the title of Figure 3 should be changed as the data presented is not related to effects of ischemia.

2) In the discussion of the angiograms presented in Figure 4, the authors should discuss whether venule perfusion returned to baseline after recanalization. Presumably, if there is hypoperfusion in capillary beds, the difference should also be seen in the exiting venules.

3) In the text associated with Figure 5, the authors mention a difference between junctional and non-junctional pericytes. This difference should be quantified.

4) In Figure 9, there should be data for at least some of the treatments showing the effect of treatment in the absence of ischemia as well as the effect of the drugs in the recanalized tissue.

5) Also in Figure 9, the increase in gap junctions should be quantified as this is an important piece of data.

6) It should be discussed in the text that phalloidin treatment might also be affecting other cell-types and what consequences that might have in the context of these experiments.

7) In the supplementary materials, there are videos nicely showing post-ischemia arterioles and pre- and post- ischemia capillaries. However, there is no video showing pre-ischemia arterioles. This is an important control so that the reader can visualize the decrease in capillary blood flow in comparison to any change in arteriole flow.

8) The convincingness of the data would be greatly increased by videos showing live contraction of pericytes on capillaries rather than relying on measuring changes in pericyte shape. It is plausible that pericyte contractions on low order vessels change blood flow, a change that is perpetuated into small vessels and causes changes in vessel diameter and pericyte shape. This caveat should at least be addressed in the text.

*Reviewer #3*:

This paper addresses several important issues related to the regulation of blood flow in the CNS. The authors demonstrate conclusively that pericytes surrounding capillaries in the retina contain α-smooth muscle actin and that this actin is responsible for pericyte contraction. This resolves an ongoing controversy, demonstrating that pericytes have the necessary molecular machinery to power vessel constriction. The authors also demonstrate a novel mechanism responsible for contraction of pericytes and constriction of capillaries following ischemia.

The authors' findings represent an important advance in our understanding of the mechanisms responsible for blood flow regulation in the healthy and pathological CNS. The paper is well written and the conclusions justified (but see below).

Subsection “Mechanisms of ischemia-induced pericyte contraction and pharmacological rescue”. The GCaMP6 demonstration of increased Ca^2+^ levels in pericytes following ischemia lacks quantification. Please include a quantitative analysis of GCaMP intensity.

Figure 9N-Q. Although the imaging results presented are suggestive, they do not establish that ischemia leads to an increased gap junctional coupling between Müller cells and pericytes. The best way to establish coupling would be to show dye coupling between the two cell types using patch pipettes. Without such evidence, the authors must state that their results suggest but do not establish such coupling.

The legend for Figure 3 is misleading. The data presented in the figure have nothing to do with ischemia-induced vessel constriction.

Figure 6. The authors state that areas of capillary constriction can be detected because blood flow is stalled in these areas. It is not clear to this reviewer how blood flow can be stalled in areas of constriction but still flow in adjacent segments of the same capillary. Flow should be uniform along the entire length of a capillary at any one time.

Figure 7. Please state what the red and green labels represent.

[Editors’ note: what now follows is the decision letter after the authors submitted for further consideration.]

Thank you for resubmitting your work entitled "Capillary Pericytes Express α-Smooth Muscle Actin and Requires Prevention of F-Actin Depolymerization for Detection" for further consideration at *eLife*. Your revised article has been favorably evaluated by Gary Westbrook as Senior editor and three reviewers.

This manuscript addresses an important issue related to the regulation of blood flow in the CNS. The authors demonstrate conclusively that pericytes surrounding capillaries in the retina contain α-smooth muscle actin. This finding resolves an ongoing controversy, demonstrating that pericytes have the necessary molecular machinery to power vessel constriction. The authors' findings represent an important advance in our understanding of the mechanisms responsible for blood flow regulation in the CNS. The paper is well written and the conclusions justified.

We ask that you address one minor issue before the paper is accepted for publication. Specifically, the authors largely discuss the total number of pericytes that are SMA+. In the discussion they also say >50% are positive. For each of the analyses they have done, in addition to give the total number of SMA+ pericytes, it would also be good to give the% pericytes that are positive for SMA

---

## [Author Response]

[Editors’ note: the author responses to the first round of peer review follow.]

Reviewer #1:[…] Major pointsThe authors assume that α-SMA is responsible for pericyte contraction, and it is a major result that using fast methanol fixation they can detect it in pericytes on high branching order capillaries. When the same result is obtained by inhibiting actin depolymerization with phalloidin or jasplakinolide the result is less convincing because these agents may be perturbing an equilibrium between the different types of actin, allowing the formation of a significant amount of F-actin that does not usually exist. This should at least be discussed.Furthermore, images need to be shown with the primary α-SMA antibody omitted (they are mentioned at the start of the Results section but not shown). Does the antibody cross react with γ-actin, which Grant & Shih (www.ncbi.nlm.nih.gov/pubmed/28933255) have suggested might mediate pericyte contraction?

We thank the reviewer for his/her constructive comments that gave us the opportunity to improve our manuscript, which we have now shortened to focus on only the α-SMA detection part.

We stained methanol-fixed as well as phalloidin pretreated retinal sections with an antibody selective for γ-actin. Most of the γ-actin immunostaining was longitudinally running parallel to the pericyte plasma membrane unlike the complete or incomplete circular staining of α-SMA. We did not observe any redistribution of the immunostaining patterns of the two isoforms after inhibiting F-actin depolymerization with phalloidin (please, see Figure 4 and the second paragraph in the Results).

We also added a supplemental figure (Figure 2—figure supplement 1) illustrating a retina when primary α-SMA antibody was omitted

Figure 3. Can the effect of the siRNA expression on SMA expression be quantified as a function of capillary branch order?

We now illustrate the effect of siRNA on α-SMA protein synthesis in pericytes based on the branch order (please, see Figure 6 and subsection “Short interfering RNA suppresses α-SMA expression in capillary pericytes**”** in the Results). After siRNA, we could detect almost no α-SMA in 6th and 7th order branches, suggesting that the smaller pool of α-SMA in these pericytes is more vulnerable to RNA interference.

Reviewer #2:[…] 1) There is no good evidence in the literature that the siRNA technique used in Figure 3 works. While a labeled siRNA would be more ideal, the knockdown should at least be quantified. Additionally, the title of Figure 3 should be changed as the data presented is not related to effects of ischemia.

We thank the reviewer for his/her constructive comments that gave us the opportunity to improve our manuscript, which we have now shortened to focus on only the α-SMA detection part.

We now provide Figure 6, illustrating the distribution of the fluorescent-tagged siRNA (which is commercially available unlike α-SMA siRNA) in the retina 48 hours after intravitreal injection. We also provide Western blot and rtPCR data comparing α-SMA expression in retinas injected with scrambled or α-SMA siRNA (Figure 6). There was a considerable reduction in α-SMA expression in siRNA-treated group; however, the differences did not reach to statistical significance possibly because the rich source of α-SMA in vascular smooth muscle cells of the arterioles and in pericytes of the upstream microvessels was not appreciably affected unlike the capillaries as clearly seen in the IHC staining and in figure illustrating α-SMA knockdown as a function of capillary branch order.

We have rephrased the title of Figure 3 (now, Figure 6) legend correctly. We are sorry for this mistake.

Reviewer #3:This paper addresses several important issues related to the regulation of blood flow in the CNS. The authors demonstrate conclusively that pericytes surrounding capillaries in the retina contain α-smooth muscle actin and that this actin is responsible for pericyte contraction. This resolves an ongoing controversy, demonstrating that pericytes have the necessary molecular machinery to power vessel constriction. The authors also demonstrate a novel mechanism responsible for contraction of pericytes and constriction of capillaries following ischemia.The authors' findings represent an important advance in our understanding of the mechanisms responsible for blood flow regulation in the healthy and pathological CNS. The paper is well written and the conclusions justified (but see below).The legend for Figure 3 is misleading. The data presented in the figure have nothing to do with ischemia-induced vessel constriction.

We thank the reviewer for his very constructive comments. We have now shortened the manuscript to focus on only the α-SMA detection part.

We are sorry for this mistake. We have rephrased the title of Figure 3 (now, Figure 6) legend correctly.

[Editors' note: the author responses to the re-review follow.]

This manuscript addresses an important issue related to the regulation of blood flow in the CNS. The authors demonstrate conclusively that pericytes surrounding capillaries in the retina contain α-muscle actin. This finding resolves an ongoing controversy, demonstrating that pericytes have the necessary molecular machinery to power vessel constriction. The authors' findings represent an important advance in our understanding of the mechanisms responsible for blood flow regulation in the CNS. The paper is well written and the conclusions justified.We ask that you address one minor issue before the paper is accepted for publication. Specifically, the authors largely discuss the total number of pericytes that are SMA+. In the discussion they also say >50% are positive. For each of the analyses they have done, in addition to give the total number of SMA+ pericytes, it would also be good to give the% pericytes that are positive for SMA

We have updated the manuscript as follows:

– We have added a new panel to Figure 3, showing the percentage of α-SMA positive pericytes for each retinal plexus (panel D).

– We did not have plexus data after jasplakinolide fixation before. We now have added absolute values of the number of α-SMA positive pericytes for each plexus for retinas treated with jasplakinolide to Figure 3. We have updated the main text, adding the new results with P values (second paragraph of subsection “Prevention of F-actin depolymerization in vivo allows α-SMA detection in capillary pericytes”)

– We have updated the legend of the Figure 3 accordingly.

– We have updated the figure citations in the main text.

– We have updated Supplementary file 1.